# *FOXE1* Gene Dosage Affects Thyroid Cancer Histology and Differentiation In Vivo

**DOI:** 10.3390/ijms22010025

**Published:** 2020-12-22

**Authors:** Sara C. Credendino, Carmen Moccia, Elena Amendola, Giuliana D’Avino, Luigi Di Guida, Eduardo Clery, Adelaide Greco, Claudio Bellevicine, Arturo Brunetti, Mario De Felice, Gabriella De Vita

**Affiliations:** 1Department of Molecular Medicine and Medical Biotechnology, University of Naples Federico II, via Pansini 5, 80131 Naples, Italy; saracarmela.credendino@unina.it (S.C.C.); Carmen.moccia26@gmail.com (C.M.); giul.davino@gmail.com (G.D.); luigi.diguida@unina.it (L.D.G.); mario.defelice@unina.it (M.D.F.); 2Institute of Experimental Endocrinology and Oncology “G. Salvatore”, National Research Council (CNR), via Pansini, 5, 80131 Naples, Italy; elena.amendola@unina.it; 3Department of Public Health, University of Naples Federico II, via Pansini 5, 80131 Naples, Italy; eduardoclery87@gmail.com (E.C.); claudio.bellevicine@unina.it (C.B.); 4Department of Advanced Biomedical Sciences, University of Naples Federico II, via Pansini 5, 80131 Naples, Italy; adelaide.greco@unina.it (A.G.); arturo.brunetti@unina.it (A.B.); 5CEINGE Advanced Biotechnology S.C.AR.L., via Gaetano Salvatore 486, 80145 Naples, Italy

**Keywords:** *FOXE1*, thyroid cancer susceptibility, mouse model, differentiation, proliferation

## Abstract

The transcription factor Forkhead box E1 (*FOXE1*) is a key player in thyroid development and function and has been identified by genome-wide association studies as a susceptibility gene for papillary thyroid cancer. Several cancer-associated polymorphisms fall into gene regulatory regions and are likely to affect *FOXE1* expression levels. However, the possibility that changes in *FOXE1* expression modulate thyroid cancer development has not been investigated. Here, we describe the effects of *FOXE1* gene dosage reduction on cancer phenotype in vivo. Mice heterozygous for *FOXE1* null allele (*FOXE1*^+/−^) were crossed with a *BRAF*^V600E^-inducible cancer model to develop thyroid cancer in either a *FOXE1*^+/+^ or *FOXE1*^+/−^ genetic background. In *FOXE1*^+/+^ mice, cancer histological features are quite similar to that of human high-grade papillary thyroid carcinomas, while cancers developed with reduced *FOXE1* gene dosage maintain morphological features resembling less malignant thyroid cancers, showing reduced proliferation index and increased apoptosis as well. Such cancers, however, appear severely undifferentiated, indicating that *FOXE1* levels affect thyroid differentiation during neoplastic transformation. These results show that *FOXE1* dosage exerts pleiotropic effects on thyroid cancer phenotype by affecting histology and regulating key markers of tumor differentiation and progression, thus suggesting the possibility that *FOXE1* could behave as lineage-specific oncogene in follicular cell-derived thyroid cancer.

## 1. Introduction

Thyroid cancer is one of the most common endocrine malignancies [1]. Papillary thyroid carcinoma (PTC) is the most frequent among the different thyroid cancer histotypes. It originates from thyroid follicular cells and accounts for 80% of all thyroid neoplasia cases [2,3]. Notably, among cancers with no Mendelian inheritance, PTC shows the highest relative risk in the first-degree relatives of probands. Despite this evidence indicating that PTC has a strong genetic component, the genes involved in PTC predisposition are poorly characterized [4]. Recently, FOXE1, a transcription factor expressed in the thyroid since early developmental stages and throughout adult life, has been associated with thyroid cancer susceptibility [5,6]. The A allele of single nucleotide polymorphism (SNP) rs965513 close to the *FOXE1* genomic locus is highly associated with increased PTC risk [7]. This is a lead SNP falling within a linkage disequilibrium block [chromosome 9: 100486812–100566532 (hg19)] that contains other co-inherited SNPs acting, in turn, as functional variants of three long-range *FOXE1* enhancers, affecting *FOXE1* expression [8]. Several other polymorphisms along the *FOXE1* gene, or in its genomic region, have been associated with both increased susceptibility to PTC and more aggressive tumors in different case-control studies, including an investigation on Chernobyl accident-derived thyroid carcinoma [9,10,11]. In particular, the rs1867277 SNP, located in the *FOXE1* promoter region, has been positively associated with the severity of histopathological features and lymph node invasion [10,12]. Landa et al. demonstrated that the two alleles of SNP rs1867277 regulate the *FOXE1* transcription rate by differentially recruiting USF1/USF2 transcription factors [12]. Beyond genomic variants, different *FOXE1* expression levels have been associated with several clinical and pathological cancer parameters, such as extra-capsular invasion, lymph node metastasis, and staging [13]. Recently, *FOXE1* expression levels have been positively associated with thyroid cancer differentiation degree, however the same study shows that silencing of *FOXE1 in vitro* decreases migratory and the invasive ability of thyroid cells [14]. Overall, the landscape emerging from the literature strongly indicates that either due to genetic variants of regulatory elements or to still unidentified trans-acting factors, the expression level of Foxe1 is the likely determinant underlying different susceptibility to PTC and/or different cancer phenotypes. Despite these evidences, the demonstration of a cause-effect relationship between *FOXE1* expression levels and thyroid cancer phenotype in vivo is still missing. 

Here, we investigate if *FOXE1* gene dosage is able to influence the outcome of experimental thyroid carcinoma in vivo by using a *BRAF* oncogene-induced thyroid cancer model with heterozygous Foxe1 knockout allele. We show that genetically reduced Foxe1 levels impact on tumor histology, proliferation, and differentiation. Our data indicate that the abundance of this transcription factor is relevant for determining thyroid cancer features, thus suggesting that *FOXE1* could play a dual role in thyroid follicular cells as it is absolutely required for normal gland development, while also being able to affect several aspects of thyroid cancer phenotype, possibly acting as a lineage-specific oncogene.

## 2. Results

### 2.1. Generation of FOXE1 Heterozygous Knockout Mouse Model of Thyroid Cancer

To investigate the role of *FOXE1* gene dosage in thyroid cancer development, we used the double transgenic Tg-rtTA/TetO*BRAF*^V600E^ mouse model (*BRAF*) bearing the reverse tetracycline transactivator (rtTA) under the control of the thyroglobulin (Tg) promoter and the *BRAF*^V600E^ oncogene under the control of a modified tetracycline operator (TetO). Doxycycline (Dox) induces the expression of *BRAF*^V600E^ (Figure 1A), leading to the rapid development of thyroid tumors resembling high-grade PTC [15]. Reduction of *FOXE1* expression levels was obtained by crossing *BRAF* mice with mice heterozygous for *FOXE1* null allele (*FOXE1*^+/−^) as homozygous *FOXE1*^−/−^ are not viable [16]. The crossing generates a total of eight different genotypes, as summarized in Figure 1B. Further analyses were performed on the two genotypes of interest, *BRAF FOXE1*^+/+^ and *BRAF FOXE1*^+/−^ mice, both double transgenic and with bi- or mono-allelic *FOXE1* expression, respectively, and *FOXE1*^+/+^ (WT) and *FOXE1*^+/−^ as controls. Representative genotype analysis of selected mice is shown in Figure 1C. Reduction of *FOXE1* gene expression in *FOXE1*^+/−^ litters was confirmed by quantitative RT-PCR. It is worth noting that although *BRAF FOXE1*^+/+^ mice show a reduction (not statistically significant) in *FOXE1* expression with respect to WT, *FOXE1* levels are lower in each *FOXE1*^+/−^ mouse line with respect to the corresponding *FOXE1*^+/+^ control line (Figure 1D). In the absence of Dox treatment, thyroids belonging to mice of different genotypes are indistinguishable by Hematoxylin and Eosin (H&E) staining (Appendix A).

### 2.2. Thyroid Cancer Histology in BRAF FOXE1 ^+/−^ Mice

Mice were divided into experimental groups, depending on the genotype and the treatment they underwent, each consisting of six animals (Appendix A). Dox treatments were performed for one week as described in methods on WT, *FOXE1*^+/−^, *BRAF FOXE1*^+/+^, and *BRAF FOXE1*^+/−^, then thyroids were dissected and analyzed by H&E staining. As expected, normal thyroid histology in both WT and *FOXE1*^+/−^ control mice was not affected by doxycycline treatment (Appendix A). Both *BRAF FOXE1*^+/+^ and *BRAF FOXE1*^+/−^ mice instead developed thyroid cancers, which show different morphologies. Specifically, *BRAF FOXE1*^+/+^ thyroid tissue displays a dramatic loss of the normal follicular structure, featuring a solid growth pattern as expected for this cancer model [15], while *BRAF FOXE1*^+/−^ glands show empty structures, resembling residual follicle lumens (Figure 2A). Follicle-like structures were counted in seven fields for each of the six analyzed thyroids for each genotype, confirming that such structures are significantly more represented in thyroid cancers of *BRAF FOXE1*^+/−^ mice with respect to that of *BRAF FOXE1*^+/+^ mice (Figure 2B).

### 2.3. Thyroid Cancer Cell Growth Suppression and Apoptosis in BRAF FOXE1^+/−^ Mice

The different cancer histology was associated with a marked reduction of cell proliferation of *BRAF FOXE1*^+/−^ cancers, as determined by immunohistochemistry (IHC) for Ki-67, compared to *BRAF FOXE1*^+/+^ (Figure 3A,B). Given the established role of FOXE1 in the survival of developing thyroid follicular cells [16], we also measured apoptosis in thyroid cancers by cleaved caspase-3 staining. As shown in Figure 3C,D, *BRAF FOXE1*^+/−^ cancers show increased apoptosis, as measured by cleaved caspase-3 positive cells number. Both these markers are not detectable in WT and *FOXE1*^+/+^ treated thyroid (Appendix A).

### 2.4. Differentiated Thyroid Gene Expression in BRAF FOXE1^+/−^ Cancers

Differentiation of *BRAF FOXE1*^+/+^ and *BRAF FOXE1*^+/−^ cancers was analyzed by IHC for two representative differentiated thyroid markers, the transcription factor PAX8, and the precursor of thyroid hormones thyroglobulin (TG). IHC analysis demonstrates that the solid structures observed by H&E staining in *FOXE1*^+/+^ thyroid cancers are composed of PAX8- and TG-positive cell nests distributed throughout the gland (Figure 4A,B left panels). Conversely, *FOXE1*^+/−^ cancers show a weaker staining for both PAX8 and TG, with the positive cells surrounding and partially filling the follicle-like structures (Figure 4A,B right panels). Furthermore, TG staining revealed that the differentiated cells retain thyroglobulin in the cytoplasm in both *FOXE1*^+/+^ and *FOXE1*^+/−^ cancers. Indeed, the follicle-like structures observed in *FOXE1*^+/−^ samples appear to be empty, thus indicating that TG is not released in the lumen. As a control, we analyzed the functional normal WT and *FOXE1*^+/−^ thyroids, after doxycycline treatment. In both WT and *FOXE1*^+/−^ follicular cells are PAX8-positive and TG-negative, with the follicle lumens containing TG-positive colloid (Appendix A). 

To further characterize the differentiation state of *BRAF FOXE1*^+/+^ and *BRAF FOXE1*^+/−^ cancers, we analyzed the expression of a panel of thyroid differentiation markers by quantitative RT-PCR. The analysis revealed that in *BRAF FOXE1*^+/−^ cancers, the decrease of all the analyzed thyroid differentiation markers is significantly more pronounced than that observed in *BRAF FOXE1*^+/+^ ones (Figure 4C), showing that lower *FOXE1* levels are associated with less differentiated tumors. Conversely, the same analysis performed in control WT and *FOXE1*^+/−^ mice confirms that the overall differentiation of *FOXE1*^+/−^ thyroids is not reduced compared to the WT ones (Appendix A).

## 3. Discussion

The transcription factor *FOXE1* was originally identified as a key player in thyroid development and function, being required since the early stages of thyroid organogenesis for migration and survival of thyrocyte precursors and in the adult thyroid for differentiation maintenance [16,17]. More recently, genome-wide association studies have highlighted a novel role of *FOXE1* as a susceptibility gene for thyroid cancer by identifying several non-coding SNPs linked to increased risk of developing PTC [5,6,7]. Interestingly, most of the identified SNPs fall into putative regulatory regions of the *FOXE1* locus, suggesting that different susceptibilities to thyroid cancer could be determined by variable expression level of *FOXE1* [8,9,12]. This hypothesis is strongly supported by large datasets available online, in which it is reported that *FOXE1* expression is higher in normal thyroid with respect to cancer (http://gepia.cancer-pku.cn/detail.php?gene=&clicktag=boxplot) and that higher *FOXE1* expression correlates with longer overall survival (https://www.proteinatlas.org/ENSG00000178919-FOXE1/pathology/thyroid+cancer).

To establish if different amounts of *FOXE1* could result in different outcomes in thyroid tumorigenesis, we generated a mouse model of thyroid cancer harboring only one functional *FOXE1* allele by crossing the heterozygous *FOXE1* knockout mouse line [16] with a well-established model of thyroid cancer, in which a *BRAF*^V600E^ oncogene is expressed in a thyroid-specific and Dox-inducible manner [15]. In the latter, thyroid tumors develop with high penetrance in both thyroid lobes within a week of Dox induction, showing histological and molecular features that closely resemble human high-grade PTC. Here, we show that decreased *FOXE1* levels exert pleiotropic effects on thyroid cancer phenotype in vivo. Histological analyses show that reduction of *FOXE1* dosage results in the loss of the expected solid pattern of tumor growth [15], with the presence of empty structures resembling follicle remnants, partially filled with cells expressing thyroid markers. The different histology is accompanied by decreased proliferation and increased apoptosis indexes. Interestingly, a similar pattern was observed in the original *BRAF* mouse model during the recovery of normal thyroid morphology after Dox withdrawal [15]. One possible explanation for this similarity is that *FOXE1* reduction in the normal thyroid could hamper the oncogenic process elicited by oncogene activation, likely by decreasing the ability of thyroid cells to survive during transformation. Indeed, the reduced proliferation index and increased apoptosis suggest that *FOXE1^+/−^* cancers display a less malignant phenotype. 

Besides the morphological differences, we highlighted that reduced *FOXE1* function induces a more pronounced loss of follicular cell differentiation, a feature commonly associated with more aggressive thyroid cancers. Such dual oncogenic-anti oncogenic activity of *FOXE1* has been recently highlighted by a paper showing that in human thyroid cancers, *FOXE1* expression positively correlates with differentiation degree, but also that *FOXE1* is able to induce *in vitro* cell migration and epithelial-to-mesenchymal transition [14]. According to these evidences, our in vivo model shows that lower *FOXE1* expression leads to the development of less differentiated cancers, which display lower proliferation and higher apoptosis rates, thereby strengthening the possibility that *FOXE1* could exert contrasting roles in thyroid neoplastic transformation. Based on these evidences, we might speculate that *FOXE1* could act as a lineage-specific oncogene [18,19,20,21], meaning that it governs lineage proliferation and survival during development as well as promoting oncogenic phenotypes in cells of the same lineage during carcinogenesis.

In conclusion, our data show for the first time in vivo a cause-effect relationship between *FOXE1* loss of function and thyroid cancer features, thus supporting the hypothesis that differential expression of the transcription factor could be the key determinant underlying its role as a susceptibility gene for differentiated thyroid cancer.

## 4. Materials and Methods 

### 4.1. Mice

TetO-*BRAF*^V600E^ and Tg-rtTA transgenic mice were obtained from Prof. J.A. Fagin at Memorial Sloan-Kettering Cancer Center (New York, NY, USA), where they were generated and were crossed to obtain the double transgenic mice Tg-rtTA; TetO-*BRAF*^V600E^. The mice were maintained under pathogen-free conditions and controlled temperature, humidity, and light and were supplied with standard or implemented food and water ad libitum in the Italian Ministry-approved (D.M. 78/213-A, and 12/2018-UT) Animal Facility of the University Federico II (Naples, Italy). Tg-rtTA-TetO-*BRAF*^V600E^ were crossed with *FOXE1* heterozygous mice to obtain *FOXE1* wt and *FOXE1* heterozygous transgenic mice, required for the study, which were fed with a 2500 mg/kg doxycycline supplemented fodder for one week. The present study was approved by the Institutional Animal Care and Use Committee (IACUC) of University of Naples Federico II and by the Italian Ministry of Health with protocol no. 2013/0078506.

### 4.2. Genotyping

Genomic DNA (gDNA) was extracted and isolated from mouse tails according to the following protocol: 400 µg of Proteinase K was used in 750 µL of lysis buffer (50 mM Tris-HCl pH8, 100 mM EDTA pH8, 100 mM NaCl, 1% SDS) and added to the samples, which were shaken for 2 h at 60 °C, 250 µL of 6 M NaCl was added, 10’ 16,000× *g* at 4 °C, 500 µL of isopropanol was added to the supernatant, 10’ 16,000× *g* at 4 °C, and the pellet was washed with 500 µL of 70% ethanol and dissolved in TE (10 mM Tris-HCl, 0.1 mM EDTA). Then, 500 ng of gDNA was used for PCR amplification performed with MyTaq^TM^ Red Mix (BIO-25044, BIOLINE, Memphis, TN, USA) according to manufacturer’s specifications.

### 4.3. Quantitative Real-Time PCR 

Total RNA was isolated from frozen mouse thyroids using Trizol (Sigma Aldrich, St. Louis, MO, USA) reagent and total cDNA was generated with SensiFAST cDNA Synthesis Kit (BIO-65054, BIOLINE, Memphis, TN, USA), according to manufacturer’s specifications. Quantitative Real-Time PCR on total cDNA was performed with Xpert Fast SYBR (Uni) (GE20, Grisp, Porto, Portugal) using gene specific oligos. *FOXE1, PAX8, TG, NIS, TSHR*, and *TPO* oligonucleotides are reported in Credendino et al. [22]. Normalization was performed by amplifying β-actin with specific oligonucleotides (Fw: 5′-ctgaaccctaaggccaaccgtg-3′; Rev: 5′-ggcatacagggacagcacagcc-3′).

### 4.4. Staining and Immunohistochemistry

Thyroids were fixed in 4% PFA, dehydrated, and paraffin embedded, as already described [23]. Then, 7 μm sections were obtained, deparaffinized, and rehydrated. The sections were permeabilized with 5′ PBS-0.2% triton and washed 2× 5′ PBS. For Hematoxylin/eosin staining, the slides were then stained with hematoxylin (Surgipath 3801562E, Leica, Wetzlar, Germany), washed in ethanol 50%, and stained in eosin (Surgipath 3801602E, Leica, Wetzlar, Germany). For IHC, they underwent unmasking treatment in citrate buffer (0.01 M pH6) 15′ in the microwave. Endogenous peroxidases were then saturated with methanol and 1.5% oxygen peroxide and tissues were permeabilized with 5′ PBS-0.2% triton, washed 2× 5′ PBS, and blocked in blocking solution (5% Normal goat serum (S-1000, Vector Laboratories, Burlingame, CA, USA), 3% BSA, 20 mM MgCl2, 0.3% tween20 in PBS) for 1 h at room temperature. Primary antibodies were used in blocking solution overnight at 4 °C. The following antibodies were used: PAX8 [22], TG (Dako, Santa Clara, CA, USA), Ki67 (Ab16667, Abcam, Cambridge, UK), and Cleaved Caspase-3 (9661S, Cell Signaling). The sections then underwent the following protocol: 5′ PBS-0.2% triton, 2× 5′ PBS, 1 h secondary antibody (biotinylated α-rabbit IgG (H+L) Vector Laboratories BA-1000) 1:100 in blocking solution for 1 h at room temperature, 5′ PBS-0.2% triton, 2X 5′ PBS, 30′ ABC (Vector Laboratories SK-4000) RT, 5′ PBS-0.2% triton, 3× 5′ PBS, DAB substrate (SK-4100, Vector Laboratories, Burlingame, CA, USA). After the staining, the samples were then dehydrated and covered with cover glasses using D. P. X. Mountant (liquid) (GRM655, HIMEDIA Laboratories, Mumbai, India). Images were obtained using Axioskop microscope equipped with an Axiocam 105 color digital camera (Zeiss, Oberkochen, Germany). Images were processed using the Axio Vision software.

## Figures and Tables

**Figure 1 ijms-22-00025-f001:**
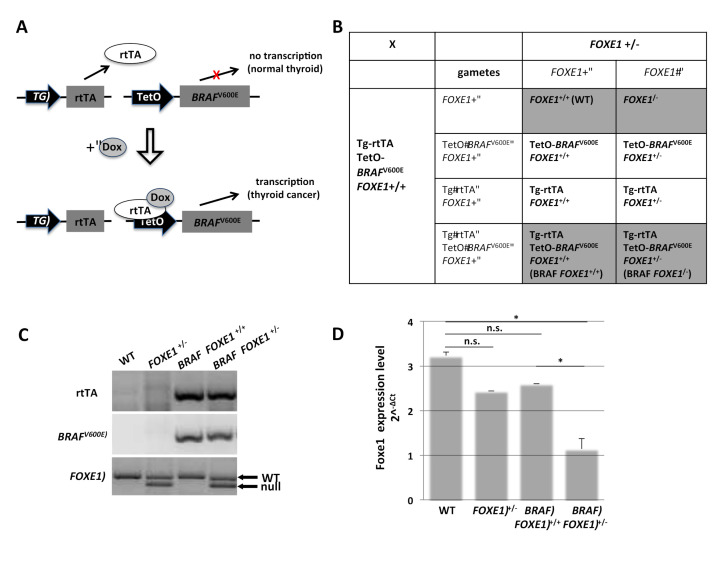
*FOXE1*^+/−^ mouse model of thyroid cancer. (**A**). Schematic representation of the Tg-rtTA/TetO*BRAF*^V600E^ double transgenic inducible mouse model of thyroid cancer (*BRAF*). The thyroglobulin (Tg) promoter drives the thyroid-specific expression of the reverse tetracycline transactivator (rtTA). In the presence of doxycycline (Dox), the rtTA is able to induce expression of a *BRAF* oncogene (*BRAF*
^V600E^) under the control of the Tet operator (TetO) and thyroid cancer develops. (**B**). Double transgenic *BRAF* mice were crossed with *FOXE1* heterozygous knock-out mice (*FOXE1*^+/−^) to obtain *BRAF* mice in both *FOXE1*^+/+^ and *FOXE1*^+/−^ genetic background. Punnet Square of the cross is shown, with genotypes used in the study shaded in gray. (**C**). Offspring were genotyped by PCR for the two transgenes rtTA and *BRAF* and for *FOXE1* wild type (WT) and knock-out (null) alleles. Representative wild type (WT), *FOXE1*^+/−^, and *BRAF FOXE1*^+/+^ or *FOXE1*^+/−^ mice genotypes are shown. (**D**). *FOXE1* levels were evaluated by quantitative RT-PCR analysis on total RNA extracted from pools of six thyroids from each genotype: WT, *FOXE1*^+/−^, *BRAF FOXE1*^+/+^, and *BRAF FOXE1*^+/−^. Data are representative of three different experiments and are reported as actin-normalized (2^−∆C_t_) values. Means ± SD are shown. * *p* < 0.05, n.s. not significant.

**Figure 2 ijms-22-00025-f002:**
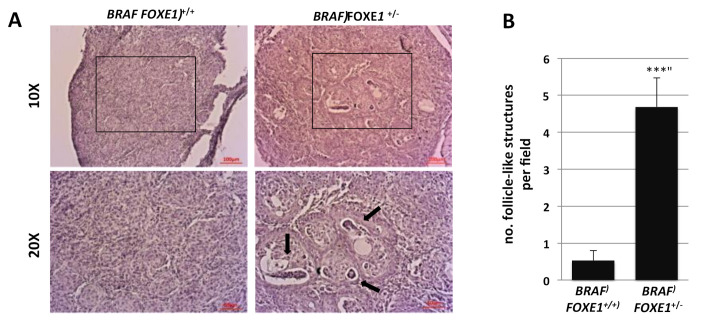
Thyroid cancer in *FOXE1*^+/−^ mice shows follicle-like structures. *BRAF FOXE1*^+/+^ and *BRAF FOXE1*^+/−^ mice were treated with doxycycline for one week to induce thyroid cancer. (**A**). Hematoxylin and eosin staining was performed on 7 µm sections. Here, 10× (upper panel) and 20× of the boxed area (lower panel) magnifications are shown. Black arrows indicate residual follicle-like structures. Each image is representative of six different images. (**B**). Follicle-like structures were manually counted in 7 fields per mouse (*n* = 6 mice) and are reported as means ± SD. *** *p* < 0.001.

**Figure 3 ijms-22-00025-f003:**
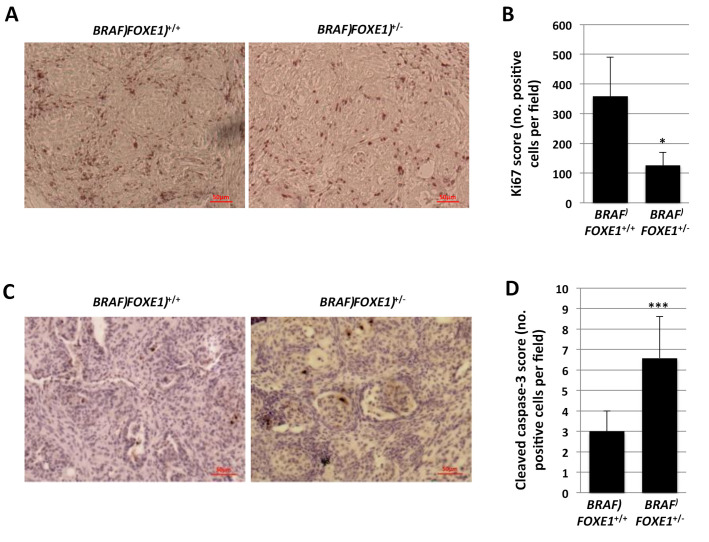
*FOXE1*^+/−^ thyroid cancer shows reduced proliferation and increased apoptosis. *BRAF FOXE1*^+/+^ and *BRAF FOXE1*^+/−^ mice were treated with doxycycline for one week to induce thyroid cancer. (**A**). IHC staining for Ki67 was performed on 7 µm sections. Here, 20× magnifications are shown. Each image is representative of six different images. (**B**). Ki67 positive cells were manually counted in four fields per mouse (*n* = 6 mice). Means of positive cells per field ± SD are reported. (**C**). IHC staining for Cleaved Caspase-3 was performed on 7 µm sections. Magnifications of 20× are shown. Each image is representative of six different images. (**D**). Cleaved Caspase-3 positive cells were manually counted in four fields per mouse (*n* = 6 mice)**.** Means of positive cells per field ± SD are reported. * *p* < 0.05, *** *p* < 0.001.

**Figure 4 ijms-22-00025-f004:**
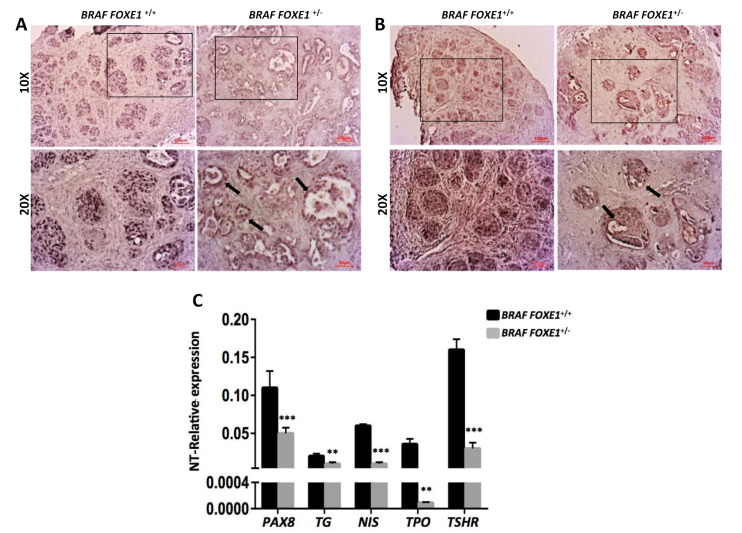
*FOXE1*^+/−^ thyroid cancer shows decreased expression of differentiation markers. IHC staining was performed on 7 µm sections from *BRAF FOXE1^+/+^* and *BRAF FOXE1*^+/−^ mice after one week of doxycycline treatment. (**A**). PAX8 and (**B**). TG. 10X (upper panels) and 20X of the boxed area (lower panels) magnifications are shown. Arrows indicate residual follicular-like structures. Each image is representative of six different images. (**C**). A panel of differentiation markers was analyzed by quantitative RT-PCR on total RNA extracted from pools of six thyroids form each experimental group: *BRAF FOXE1*^+/+^ and *BRAF FOXE1*^+/−^ mice, either Dox-treated or untreated. Data are normalized by actin expression and reported as the fold change (2^-∆∆C_t_) of treated mice with respect to untreated mice with the same genotype. Means ± SD are shown. ** *p* < 0.01, *** *p* < 0.001.

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
