# Peer review of "FOXE1 Gene Dosage Affects Thyroid Cancer Histology and Differentiation In Vivo"

_ijms, 2020, doi:10.3390/ijms22010025_

Round 1

Reviewer 1 Report

The Authors present interesting data on the role of FOXE1 expression in the development of thyorid cancer using mouse models. The data presented are important to understand the possible genetic influence of FOXE1 in thyroid cancer. However some issues arise formt he presented data, in particular the qRT-PCr for FOXE1 levels is not quite clear, in particular it is difficult to understand why the wt has such high levels of expression comprared to the foxe1 +/+ model. Moreover it is not clear what type of samples was used for normalization of the data. This is true also for the expression of thyroid-specific genes (ie. pax8 etc). These experiments need to be carefully evaluated and statistical analysis reported for the foxe1 expression.

Author Response

We thank the Reviewer for constructive criticism that helped us to greatly improve our paper.

  • (...) the qRT-PCr for FOXE1 levels is not quite clear, in particular it is difficult to understand why the wt has such high levels of expression comprared to the foxe1 +/+ model.

We thank the reviewer for this remark. The reduction of about 20% of FOXE1 expression observed in BRAF FOXE1+/+ compared to WT mice is likely due to the different genetic backgrounds of the two models. Nevertheless, such FOXE1 reduction is not statistically significant. We added a sentence to the results section commenting this observation.

  • (...) it is not clear what type of samples was used for normalization of the data.

We apologise for being not clear in the description of this data. Foxe1 expression values in Figure 1D are normalized by ß-actin expression and reported as 2^-∆Ct,. We added this missing information in both figure legend and Material and Methods section.

  • This is true also for the expression of thyroid-specific genes (ie. pax8 etc).

Expression values of thyroid specific genes reported in Figure 4C are normalized by ß-actin expression and reported as the fold change (2^-∆∆Ct) of treated mice respect to untreated mice, for each genotype. This information has been added to the figure legend.

  • These experiments need to be carefully evaluated and statistical analysis reported for the foxe1 expression.

To re-evaluate qRT-PCR data, statistical analyisis was performed for the charts where it was missing and added to Figure 1D and its legend.

Reviewer 2 Report

Manuscript: Foxe1 gene dosage modulates thyroid cancer histology and differentiation in vivo. Sara Carmela Credendino et al.

Authors performed the interesting in vivo experiments and proved the vital role of the transcription factor FOXE1 in thyroid carcinoma.  In my opinion, the presented research serves important data in the thyroid cancer field with clear explanations in the paper.  However, the manuscript can be improved following the minor comments and proofreading providing.

  1. Gene names should be capitalized and in italic – e.g., FOXE1, BRAF
  2. In vivo should be written in italic – e.g. line 70
  3. In the Abstract the full name of FOXE1 - Forkhead box protein E1 - should be provided
  4. The letters on the graphs should be bigger
  5. g. in Figure 2B, 3B – please make the stars closer to the graph
  6. I think according to the rules of the journal you should write the full word “Figure” in the text
  7. Figure 4C – please make the bigger graph with results
  8. In the literature, you need to correct the double numbers
  9. Figure S1, S2 – please make better contrast, lower background on the pictures
  10. The different parts of the result section should be numbered 2.1, 2.2. etc.
  11. “One possible  explanation  for  193this   similarity   is   that   Foxe1   reduction   in   the   normal   thyroid   could   in   some   way   hamper   the  oncogenic   process   elicited   by   oncogene   activation…” – please include your speculation of the mechanism of this regulation and avoid the term “ some way”
  12. According to GEPIA and Protein Atlas database (i) FOXE1 expression level shows a higher tendency in normal tissues and (ii) its higher expression correlates with longer survival.                                 (i)http://gepia.cancer-pku.cn/detail.php?gene=&clicktag=boxplot (ii) https://www.proteinatlas.org/ENSG00000178919-FOXE1/pathology/thyroid+cancer

Please consider mentioning this information in your manuscript e.g. in the discussion

Author Response

We thank the Reviewer for constructive criticism that helped us to greatly improve our paper.

  1. Gene names should be capitalized and in italic-e.g. FOXE1, BRAF.

We thank the reviewer for the suggestion. We modified the format of gene names all along the text.

  1. In vivo should be written in italic - e.g. line 70.

We corrected this format all along the text.

  1. In the Abstract the full name of Foxe1- Forkhead box protein E1- should be provided.

We provided the full name in the abstract, where we mention the transcription factor for the first time. Thanks for the suggestion.

  1. The letters on the graphs should be bigger.

We increased the font of the letters that mark the panels and the titles of the chart axes, as suggested.

  1. In Figure 2B, 3B-please make the stars closer to the graph.

Done.

  1. I think according to the rules of the journal you should write the full word "Figure" in the text.

We replaced the abbreviation "Fig" with the full word "Figure" throughout the text.

  1. Figure 4C - please make the bigger graph with results.

We modified Figure 4 in order to enlarge graph 4C.

  1. In literature, you need to correct double numbers.

We corrected them.

  1. Figure S1, S2-please make better contrast, lower background on the pictures.

We modified the pictures as requested.

  1. The different parts of the result section should numbered 2.1, 2.2 etc.

We numbered the paragraphs of the result section according to the reviewer's request.

  1. "One possible explanation for this similarity is that Foxe1 reduction in the normal thyroid could in some way hamper the oncogenic processes elicited by oncogene activation..."- please include your speculation of the mechanism of this regulation and avoid the term "some way".

We thank the reviewer for this remark. We changed the sentence "One possible explanation for this similarity is that Foxe1 reduction in the normal thyroid could in some way hamper the oncogenic process elicited by oncogene activation, thus resulting in a less malignant cancer phenotype, as suggested also by the reduced proliferation index and increased apoptosis observed" with the following one: "One possible explanation for this similarity is that FOXE1 reduction in the normal thyroid could hamper the oncogenic process elicited by oncogene activation, likely by decreasing the ability of thyroid cells to survive during transformation. Indeed, the reduced proliferation index and increased apoptosis suggest that FOXE1+/- cancers display a less malignant phenotype".

  1. According to GEPIA and Protein Atlas database (i) FOXE1 expression level shows a higher tendency in normal tissues and (ii) its higher expression correlates with longer survival. (i) http://gepia.cancer-pku.cn/detail.php?gene=&clicktag=boxplot (ii)https://www.proteinatlas.org/ENSG00000178919-FOXE1/pathology/thyroid+cancer. Please consider mentioning this information in your manuscript e.g. in the discussion.

To receive the reviewer's suggestion, we included this information in the discussion section, by adding the following sentence: "This hypothesis is strongly supported by large datasets available online, in which it is reported that FOXE1 expression is higher in normal thyroid respect to cancer (http://gepia.cancer-pku.cn/detail.php?gene=&clicktag=boxplot), and that higher FOXE1 expression correlates with longer overall survival (https://www.proteinatlas.org/ENSG00000178919-FOXE1/pathology/thyroid+cancer)".

Round 2

Reviewer 1 Report

The Authors addressed all the major issues of the previous draft. I have no additional requests.